# Modelling of Mass and Thermal Balance and Simulation of Iron Sintering Process with Biomass

**Jaroslav Legemza \*, Mária Fröhlichová, Róbert Findorák and Martina Džupková**

Faculty of Materials, Metallurgy and Recycling, Technical University of Košice, Letná 9, 04200 Košice, Slovakia; maria.frohlichova@tuke.sk (M.F.); robert.findorak@tuke.sk (R.F.); martina.dzupkova@tuke.sk (M.D.)

**\*** Correspondence: jaroslav.legemza@tuke.sk; Tel.: +421-55-602-3155

**Abstract:** This paper specifies the mathematical and physical modelling of the iron sintering process in laboratory conditions. The aim is to get the simplest approach (using thermodynamic software "HSC Chemistry", version 9, Outokumpu Research Oy, Pori, Finland) that allows one to predict the output parameters based on the initial composition analysis. As a part of the application of mathematical modelling, a mass and thermal balance of combustion of carbonaceous fuels (including biomass) and a mass and thermal balance of high-temperature sintering of an agglomeration charge were determined. The objective of the paper was to point out the advantages of modelling using thermodynamic software and apply the results into a simulation of the sintering process. The outcome of mathematical modelling correlates to the outcome of physical modelling for fuel combustion and the agglomerate production in a laboratory sintering pan. The energy required to reach the desired sintering temperatures and acquire the standard quality of agglomerate was calculated using 4.97% of coke breeze. In a real experiment with the laboratory sintering pan, 4.35% of coke was used. When a biomass fuel with a lower calorific value (lignin) is used in the agglomeration charge, the amount of fuel has to be increased to 5.52% (with 20% substitution of coke). This paper also aimed at predicting methodological tools and defining thermodynamic conditions for creating an interactive simulation. In addition, kinetics should be considered to improve the predicting capabilities of the current model and therefore in further research it will be required to optimise the computational program pursuant to the results of the kinetics experiments.

**Keywords:** agglomerate; carbon fuel; coke breeze; biomass; lignin; modelling; mass-thermal balance; temperature; sintering

## 1. Introduction

The state of play in technology and the current requirements on the production of iron and steel call for up-to-date automation. It is also important for addressing new technological and research challenges in the field of processing ferriferous fine-grained materials. These processes require mathematical and physical modelling, a sophisticated approach, and up-to-date facilities. The monitoring, automation, and utilization of modern analytical tools is very important. It pertains to the sintering process as well, where there are many manual operations dependant on the human factor. In operating conditions, the sintering process takes place continuously on sintering belts. It is a non-linear technological processes affected by a high number of factors that are difficult to evaluate. Therefore, the individual parameters (mainly the composition of input materials, amount of fuel, temperatures in sintered layer, and pressures of sucked air) of the sintering process are mostly examined and analysed in laboratory conditions. Results from these experiments can provide verifiable conclusions for the industry [1]. In the world, the production of agglomerate in laboratory conditions is carried out

on static sintering equipment. It is necessary to state that the conditions of sintering on laboratory sintering equipment slightly differ from the conditions of sintering on a sintering belt both in terms of temperatures and gas flow. The possible application of the models in practice requires the mathematical prediction of the results of laboratory experiments using regression dependences [2]. The creation of models, simulations, and predictions is very important today. For these models we can define physical-chemical and thermal processes that alter the structure and composition of input sintering raw materials. The result is a blast furnace agglomerate with the required properties produced efficiently and environmentally.

In Slovakia, a laboratory sintering pan (LSP) was used and was fully equipped with measuring devices and analysers to sufficiently simulate the conditions on the sintering belt in relation to temperature conditions in the sintered layer, production of CO, $CO_2$, $NO_x$, $SO_x$, and particulate matter (PM) in the sintering process, as well as to the quality of the agglomerate [3].

The laboratory sintering pan was located at the Institute of Metallurgy, Faculty of Materials, Metallurgy and Recycling, Technical University of Košice (UMET FMMR TUKE) (see Figure 1). Thermodynamic and mathematical models focused primarily on the mass and thermal balance of the agglomerate production are also a part of this physical model of sintering [3].

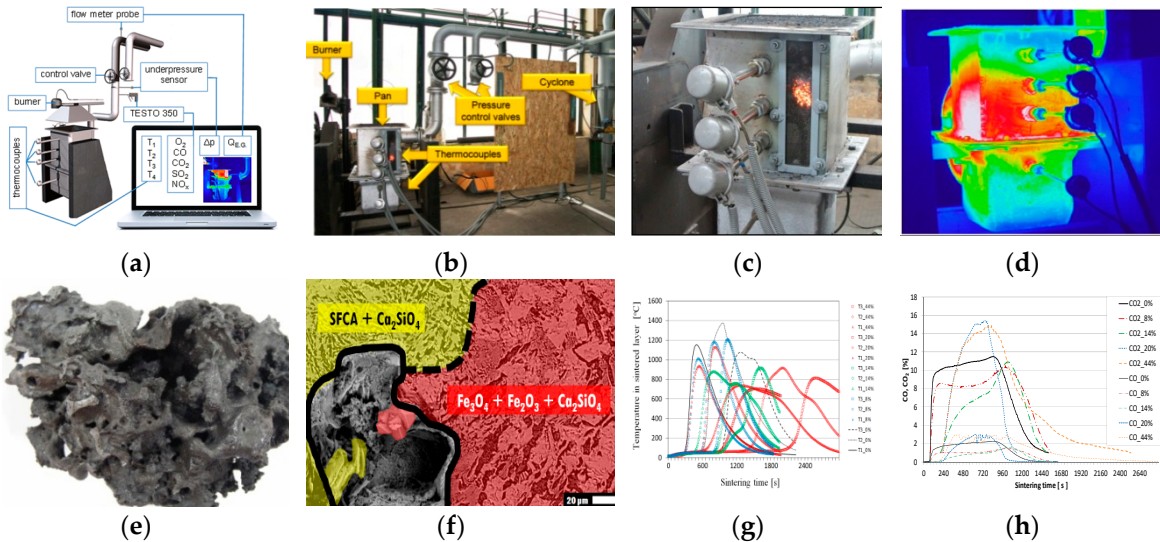

**Figure 1.** Laboratory sintering pan and modelling of agglomerate production at UMET FMMR TUKE (Košice, Slovakia)—reproduced from [3], with permission from the Technical University of Košice, Slovakia, 2015. (**a**) Illustration of the laboratory sintering pan; (**b**) real laboratory sintering pan; (**c**) sintering in the laboratory sintering pan; (**d**) infrared temperature measurement in the iron sintering process; (**e**) final agglomerate; (**f**) structure of agglomerate; (**g**) temperatures in the sintered layer; and (**h**) emission profile of the sintering process.

The sintering process takes place in a heterogeneous system of the gas–liquid–solid phase, while the gas phase ensures the fuel combustion, heat transfer, and oxidation-reduction processes. The process of sintering does not occur simultaneously across the layer. It occurs gradually in a narrow zone (the so-called combustion zone), which moves towards the grate. The heat in each elementary layer of the charge during sintering is given by the exothermic effect of burning fuel and this process is very important for the creation of sintering melt [4]. Due to the progress of the combustion zone and the major heat transfer by convection, heat accumulation occurs in the elementary layers towards the sintering grates. The static sintering of the sintering layer in Figure 2 shows the temperature distribution in the static sintered layer. After the ignition of an agglocharge, the combustion zone gradually moves downwards to the lower layers of the charge, increasing the maximum temperatures in the sintered layer [5].

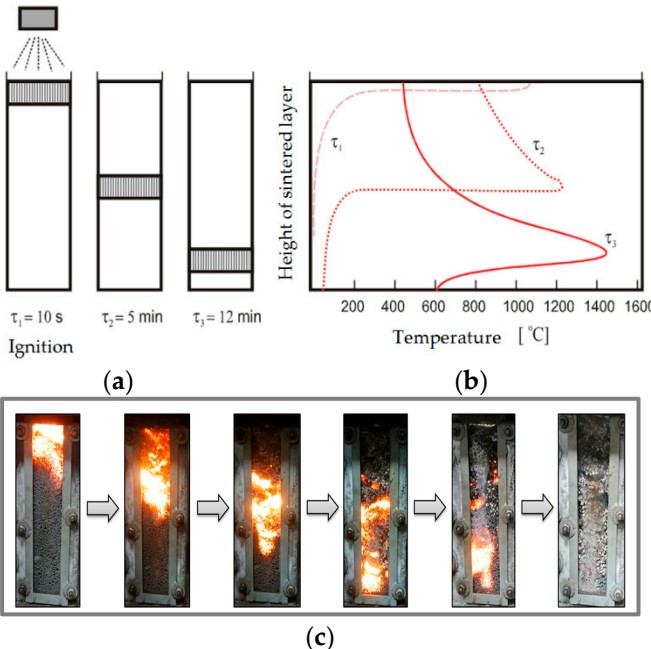

**Figure 2.** Course of sintering and temperature distribution in the static sintered layer—reproduced from [3,5]. (**a**) Progress of the combustion zone after the ignition of the charge surface at sintering times $\tau_1$–$\tau_3$; (**b**) temperatures in sintered layer at sintering times $\tau_1$–$\tau_3$; and (**c**) visible fuel burning zone in the sintering pan.

The agglomeration process has been modeled worldwide in several studies. Zhou and colleagues show the predicted melting and solidification heats and melt fraction as functions of time [6]. These processes are very important for forming the final agglomerate structure. Young developed a 1-D mathematical model which studied in detail the heat and mass transfer processes in a sintering bed [7]. Yamaoka and Kawaguchi [8] discussed 3D variations on sinter properties produced on a pot apparatus experimental facility and presented a mathematical model based on the transport phenomena to simulate the experimental conditions in the sintering process. The melt formation temperature, liquid phase content, and liquid viscosity under different sintering temperatures were simulated to study the influence of these thermodynamic melt formation characteristics on the liquid phase fluidity of iron ore in the sintering process [9]. Most modelling processes worldwide deal with heat generation and transfer, melt formation, and sinter cooling.

In the iron and steel production processes, it is very important to reduce fuel [10]. The minimization of fuel consumption in the sintering process is ensured by the mathematical regression model aimed at calculating the mass and thermal balance of the agglomerate production [6,8]. The mass and thermal balance is based on measurements. However, certain assumptions need to be considered, as the sintering process takes place not only in the thermodynamic but also in the kinetic mode [5]:

1.  With normal fuel consumption (up to 6%), $CO_2$ and CO are formed in the combustion of carbon at the ratio of $CO_2/CO = 4$. With the high fuel consumption (above 7%), the $CO_2/CO$ ratio is reduced to 2.5–3. The amount of heat produced in the combustion of carbon is:

$$C + O_{2\,(g)} = CO_{2\,(g)} \qquad (33{,}411 \text{ kJ/kg C}) \qquad (1)$$

$$C + {}^1\!/_2\, O_2\,(g) = CO_{2\,(g)} \qquad (+9797 \text{ kJ/kg C}) \qquad (2)$$

2.  The heat of agglomeration charge ignition reaches the value from 14,654 up to 16,747 kJ per 100 kg of the agglomerate.

3.  A total amount of 95% of the organic and sulphide sulphur is removed. A total ammount of 60% of the sulphate sulphur is removed. The thermal balance for the removal of sulphur is:

$$S + O_{2\,(g)} = SO_{2\,(g)} \qquad (+9278 \text{ kJ/kg S}) \tag{3}$$

$$4\,FeS + 7\,O_{2\,(g)} = 2\,Fe_2O_3 + 4\,SO_{2\,(g)} \qquad (+6962 \text{ kJ/kg FeS}) \tag{4}$$

4.  The amount of heat used to evaporate the moisture is:

$$H_2O_{\,(l)} = H_2O_{\,(g)} \,(-2258 \text{ kJ/kg } H_2O) \tag{5}$$

5.  The following volumes are used to decompose carbonates, i.e., to remove 1 kg of $CO_2$: from:

$$CaCO_3 \,(-4044 \text{ kJ/kg } CO_2) \tag{6}$$

from:
$$MgCO_3 \,(-2311 \text{ kJ/kg } CO_2) \tag{7}$$

from:
$$CaMg(CO_3)_2 \,(-3655 \text{ kJ/kg } CO_2) \tag{8}$$

from:
$$FeCO_3 \,(-1938 \text{ kJ/kg } CO_2) \tag{9}$$

6.  The amount of heat for the thermal dissociation of oxides is 18,288 kJ/kg of $O_2$ (up to the final dissociation of $Fe_2O_3$ to $FeO$).
7.  The heat of the hot final agglomerate varies between 33,494 and 60,242 kJ per 100 kg of agglomerate.

The creation of a computational model of the sintering process mass balance is based on the knowledge of processes taking place in the sintered layer, as well as the principles of conservation of mass and thermodynamic stability of compounds [11,12]. The effects of the composition of the sintering mixture on the formation of the melt phase under specified sintering conditions were modelled by Chen [11]. The volume and properties of the melt formed during sintering depend heavily on temperature, which determines the fuel requirement for sintering.

It follows that the combustion of solid fuel represents the greatest thermal effect and is used mainly for the dissociation reactions and constitutes the heat of agglomerate and flue gas [5].

Mathematical models are the basis of computational and theoretical methods of determining the parameters of sintering equipment, their operating modes, and predicting the properties of the sintering product [12,13]. The main criterion of the agglomeration process is that the quality produced of the agglomerate is with high reducibility while maintaining the ecological nature of the production [14]. A thermodynamic analysis is employed in the creation of mathematical models to define the chemical processes and study the mass and thermal balance [11].

Thermodynamic calculations are thus essential when determining the characteristics of a technological process and they enable one to clarify the formation of the major sintering products. By changing the basic condition of the thermodynamic system, it is feasible to find the optimum operating conditions of the sintering process and minimise the consumption of raw materials and energy. Table 1 shows the thermal balance of the sintering process, it follows that the greatest thermal effect to sintering layer is due to fuel combustion. For the production of agglomerate, energy from a coke can be technologically replaceable with biomass [15]. The use of biomass within the high-temperature sintering process contributes to the overall lower production of emissions compared to coke.

**Table 1.** Thermal balance of sintering process data from [5].

| | Heat Input | | Thermal Effect (kJ per 100 kg of Agglomerate) | Share (%) |
|---|---|---|---|---|
| 1 | Heat of charge ignition | [$Q_{ignit}$] | 14,654 | 6.66 |
| 2 | Heat of solid fuel combustion | [$Q_{fuel}$] | 188,194 | 85.55 |
| 3 | Calorific volume of sucked air | [$Q_{air}$] | 1637 | 0.74 |
| 4 | Calorific volume of charge | [$Q_{charge}$] | 13,862 | 6.31 |
| 5 | Heat of organic sulphur combustion | [$Q_s$] | 1633 | 0.74 |
| | Total | | 219,980 | 100 |
| | **Heat Consumption** | | | |
| 1 | Dissociation heat of charge moisture | [$Q_{evap}$] | 31,453 | 15.69 |
| 2 | Dissociation heat of charge carbonates | [$Q_{carb}$] | 55,224 | 27.56 |
| 3 | Dissociation heat of iron oxides | [$Q_{dis}$] | 2816 | 1.4 |
| 4 | Calorific volume of flue gas | [$Q_{fg}$] | 55,941 | 27.91 |
| 5 | Calorific volume of agglomerate | [$Q_{aggl}$] | 55,000 | 27.44 |
| | Total | | 200,435 | 100 |
| | Thermal losses | | | 8.88 |

The literature review shows that there are several separate models for specific monitored parameters worldwide, but they do not run online. Therefore, the development of a more comprehensive model would represent an innovative solution in the field of sintering process automation. This paper is aimed at describing the mathematical-thermodynamic model that could be used to predict the thermodynamic conditions of carbonaceous fuel combustion (including biomass) in the sintering process.

## 2. Materials and Methods

This paper proposes an application of the mathematical model and a physical simulation of the sintering process in laboratory conditions. For mathematical modelling the basic chemical reactions with standard Gibbs energy and mass and thermal balance were calculated. Thermodynamic data was obtained from the software HSC Chemistry [16] (HSC means H—enthalpy, S—entropy, C—heat capacity). HSC Chemistry offers powerful calculation methods for studying the effects of different variables on the chemical system at equilibrium. The aim is to get the simplest approach (using this software to calculate equilibrium) which allows one to predict the output parameters (amounts, chemistry, mineralogical composition, and total heat) based on the initial composition analysis. Thermochemical calculations are based on enthalpy H, entropy S, heat capacity Cp or Gibbs energy G values for chemical species. They can all be mathematically derived from experimental observations. The following text gives a brief and simplified but illustrative idea of thermochemical quantities and methods.

The absolute values of enthalpy *H* of substances cannot be measured, but enthalpy differences (d*H*) between two temperatures (d*T*) can be determined with a calorimeter. Heat capacity *Cp* at constant pressure *p* (specific heat) can be calculated from the data using Equation (10):

$$c_p = \left(\frac{dH}{dT}\right)_{P,n} \quad (J \cdot K^{-1}) \tag{10}$$

Equation (10) allows the calculation of enthalpy (11):

$$\Delta H(T) = \Delta H_f(298.15\ \text{K}) + \int_{298.15K}^{T} C_P dT + \Delta H_{tr} \quad (J) \tag{11}$$

where $\Delta H_f$ (298.15 K) is the enthalpy of formation at 298.15 K and $\Delta H_{tr}$ is the enthalpy of transformation of the species. The enthalpy of compounds also includes their enthalpy of formation $\Delta H_f$ from elements.

The thermodynamic enthalpy and Gibbs energy functions for chemical reactions used in the model are calculated as the difference between the products and reactants using Equations (12) and (13):

$$\Delta_R H^o_{(T)} = \left(\sum_{i=1}^{N} v_i \Delta H^o_{i(T)}\right)_{prod} - \left(\sum_{i=1}^{N} v_i \Delta H^o_{i(T)}\right)_{react} \quad (J) \tag{12}$$

$$\Delta_R G^o_{(T)} = \left(\sum_{i=1}^{N} v_i \Delta G^o_{i(T)}\right)_{prod} - \left(\sum_{i=1}^{N} v_i \Delta G^o_{i(T)}\right)_{react} \quad (J \cdot mol^{-1}) \tag{13}$$

where the following abbreviations have been used:

$\Delta_R H^o_{(T)}$ = the enthalpy change of reaction,

$\Delta_R G^o_{(T)}$ = the Gibbs energy change of reaction,

$\Delta H^o_{i(T)}$ = the enthalpy change of the species $i$ (product or reactant) in the temperature of $T$ (K),

$\Delta G^o_{i(T)}$ = the Gibbs energy change of the species $i$ (product or reactant) in the temperature of $T$ (K),

$v_i$ = stoichiometric coefficient of a species in reaction.

The composition of the equilibrium mixture is expressed by the initial composition and the extent of reaction (or degree of conversion), Equation (14):

$$n_{i,0} + v_i \zeta = n_i \quad (mol) \tag{14}$$

where the following abbreviations have been used:

$n_{i,0}$ = number of moles of the substance in the system before the reaction,

$v_i \zeta$ = number of moles of substance $i$ involved in the reaction, ($\zeta$ + for products, − for reactants),

$n_i$ = number of moles of the substance in equilibrium.

The thermodynamic equations from the program HSC Chemistry were used to obtain the necessary data for the respective chemical reactions taking place in the sintering process. These include fuel combustion, drying, calcination, oxido-reduction reactions, etc., (see Figure 3). All the thermochemical data required in HSC and its modules can be calculated from the basic data in its databases using Equations (10)–(14). In order to incorporate the output mineralogical composition of the agglomerate into the program, Gibbs equilibrium diagrams had to be calculated first and suitable phases for the output were predicted on the basis of stoichiometric calculations. The first step was to specify the chemical reaction system, with its phases and species, and give the amounts of raw materials (M, m, n, Nm$^3$). The program calculates the amounts of products at equilibrium in isothermal and isobaric conditions. The basic idea of the heat balance module is to specify the IN and OUT species, temperatures, and amounts and the Heat Balance module automatically calculates the heat and material balances [16]. The Heat Content is used to describe the energy which may be released when the compound is cooled down from the given temperature to 298.15 K. The enthalpies given in the Total H contain the values of the Heat Content as well as the heat of formation reactions. These values are used to calculate heat balances. In order to calculate a heat balance, one must first convert the (elemental-$X_m$) chemical analysis of the raw materials and the products into input and output substances (species). Sometimes this step is the most difficult, so one may choose to carry out this procedure using the HSC Species Converter module. It is important to check the element balance by selecting Calculate/Element Balance, in order to avoid incorrect material and heat balances [16]. Figure 3 shows the global method with modelling the mass and thermal balance of the sintering process.

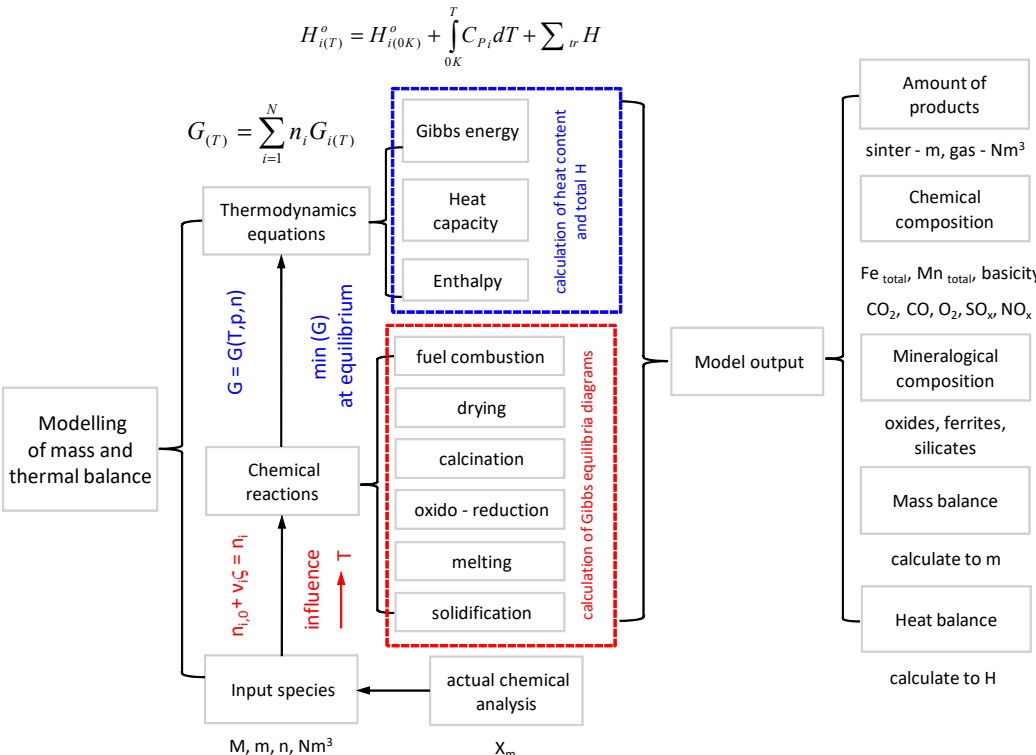

**Figure 3.** Scheme presenting the global method with modelling the mass and thermal balance.

Using the thermodynamic program HSC 9 (see Figure 4b), all outputs were calculated according to the created models. The experiments were carried out in a laboratory sintering pan (LSP), which simulates the conditions on an agglomeration belt, as well as the quality of the agglomerate (see Figures 1a and 4a). For the high temperature range in the sintered layer, three thermocouples of the PtRh10-Pt type were used. The flue gas temperature was read at two levels by the NiCr-Ni type thermocouple. The chemical composition and temperature of the flue gas were analysed by the TESTO 350 device. The differential pressure was measured by the Anubar type probe. Laboratory experiments on the laboratory sintering pan were performed using ferriferous raw materials-aggloore from the Ukraine (content of $Fe_{TOT} = 60.39\%$) and concentrate from the Ukraine (content of $Fe_{TOT} = 67.95\%$) (see Table 2). The concentrate/iron ore ratio in the mixtures was 1:1. In the agglomeration process, standard coke breeze with a grain size <3 mm was used as fuel (see Table 3). Biomass was a technical hydrolyzed lignin.

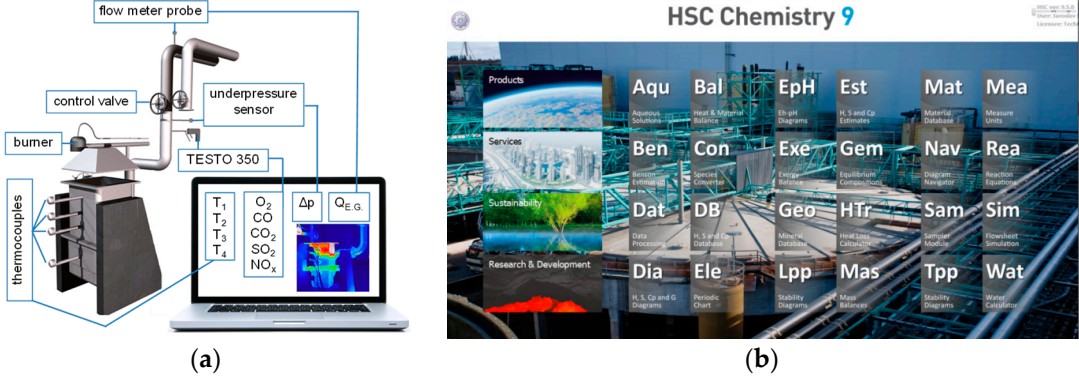

(**a**)                               (**b**)

**Figure 4.** Laboratory sintering pan (LSP) (**a**), and computational program HSC Chemistry—adapted from [16], which has a program license from HSC Chemistry, Outokumpu Research Oy, Pori, Finland, 2018 (**b**).

Lignin is a by-product of ethanol production by the distillation of wood. The lignin was subjected to an experimental study for its use as a partial replacement of the agglomeration coke for the production of an iron ore agglomerate. Details about lignin are presented in the literature [17]. The other mixture components (dolomite, calcite, and lime) had a commercial standard chemical composition. These inputs were included in prepared agglomeration mixtures with basicity within the interval of 1.5–1.9. The produced Fe agglomerates had $Fe_{TOT}$ content within the interval of about 48–53 wt%. Chemical analysis of the samples was determined using XRF spectrometer ARL 9900S (Thermo Fisher Scientific, Waltham, MA, USA). The minerals species were detected using XRD spectrometer SEIFERT XRD 3003/PTS (General Electric Company, Boston, MA, USA). Diffraction records were evaluated by DIFFRAC.EVA (Search-Match, KARLSRUHE & VIERSEN, Bruker, Billerica, MA, USA) with PDF2 and TOPAS software (version 4, Bruker, Billerica, MA, USA) using the Rietveld method.

**Table 2.** Chemical analysis of Fe materials.

| Iron Material | Humidity | Fe | FeO | $Fe_2O_3$ | $SiO_2$ | $Al_2O_3$ | CaO | MgO | Mn | P |
|---|---|---|---|---|---|---|---|---|---|---|
| | (%) | | | | | (wt%) | | | | |
| Ore | 4.20 | 60.39 | 0.52 | 85.38 | 11.07 | 0.90 | 0.07 | 0.21 | 0.027 | 0.028 |
| Concentrate | 9.79 | 67.95 | 27.80 | 66.16 | 4.89 | 0.16 | 0.12 | 0.24 | 0.030 | 0.011 |
| | **Basicity** | **S** | **$Na_2O$** | **$K_2O$** | **$TiO_2$** | **Pb** | **Zn** | **As** | **Cl** | **C** |
| | (-) | | | | | (wt%) | | | | |
| Ore | 0.024 | 0.015 | 0.154 | 0.053 | 0.038 | 0.001 | 0.001 | 0.001 | 0.210 | 0.070 |
| Concentrate | 0.070 | 0.123 | 0.029 | 0.060 | 0.015 | 0.001 | 0.001 | 0.002 | 0.050 | 0.152 |

**Table 3.** Chemical analysis and energy content of fuels.

| Fuel | Proximate Analysis (wt%) | | | | Ultimate Analysis (wt%) | | | | | |
|---|---|---|---|---|---|---|---|---|---|---|
| | $H_2O$ (W) | Ash (A) | Volatile (CV) | Fixed Carbon ($C_{FIX}$) | C | H | O | N | S | Caloric Value (MJ/kg) |
| Coke dust | 5.5 | 12.10 | 1.50 | 80.9 | 85.4 | 0.30 | 0.60 | 1.30 | 0.30 | 28.16 |
| Lignin | 8.6 | 3.4 | 67.90 | 20.1 | 62.9 | 5.75 | 27.60 | 0.2 | 0.15 | 23.14 |

## 3. Results and Discussion

### 3.1. Mathematical Modelling of Carbonaceous Fuel Combustion

As part of mathematical modelling, a thermodynamic study of coke breeze and biomass (lignin) combustion using thermodynamic program HSC Chemistry was carried out while thermodynamic models created on the basis of actual fuel composition. To calculate the thermodynamic models, Gibbs equilibrium diagrams (Equilibrium Calculations module) were utilized to characterise the temperature dependence of the equilibrium composition of considered reactants (actual fuel + air) and products (process gases and ash produced). The calculated thermodynamic modules also allowed the prediction of agglomeration charge sintering systems in the presence of coke breeze and biomass in various mixing ratios and combinations. The modelled systems and the results of thermodynamic calculations can specify the effect of the amount and type of fuel used on the oxidation-reduction processes in the temperature dependence while also being able to predict the phase compositions of ash (in case of fuel combustion) and the phase compositions of agglomerates (in case of agglomeration charge sintering). Figure 5 shows the calculated Gibbs equilibrium diagrams of coke breeze burning during its complete combustion while the majority product is $CO_{2 (g)}$ with a more than 90% share in the mixture of process gases (except for nitrogen). The amount of minority combustion products ($CO_{(g)}$, $H_2O_{(g)}$, $H_{(g)}$, $SO_{2 (g)}$) was related to the content of volatile combustible (1.5 wt%), moisture (5.5 wt%), and the content of sulphur (0.30%) in the coke breeze. Based on the calculated Gibbs equilibrium diagrams, it was feasible to predict both the majority and minority phases in ash (content = 12.1 wt%) produced by the combustion of coke breeze. The majority mineralogical phases formed at the temperatures

of coke breeze combustion include quartz ($SiO_2$), hercynite ($FeAl_2O_4$), oligoclase ($CaAl_2Si_2O_8$), and mullite ($Al_6Si_2O_{13}$). The phases of quartz and mullite were identified in X-ray diffraction as well [17]. The predicted minority mineralogical phases include fayalite ($Fe_2SiO_4$), wollastonite ($CaSiO_3$), hematite ($Fe_2O_3$), magnetite ($Fe_3O_4$), and anhydrite ($CaSO_4$). The phases of hematite and anhydrite were identified in X-ray diffraction as well [17]. Alkalis in ash are predicted in the form of silicates ($K_2SiO_3$ and $Na_2SiO_3$).

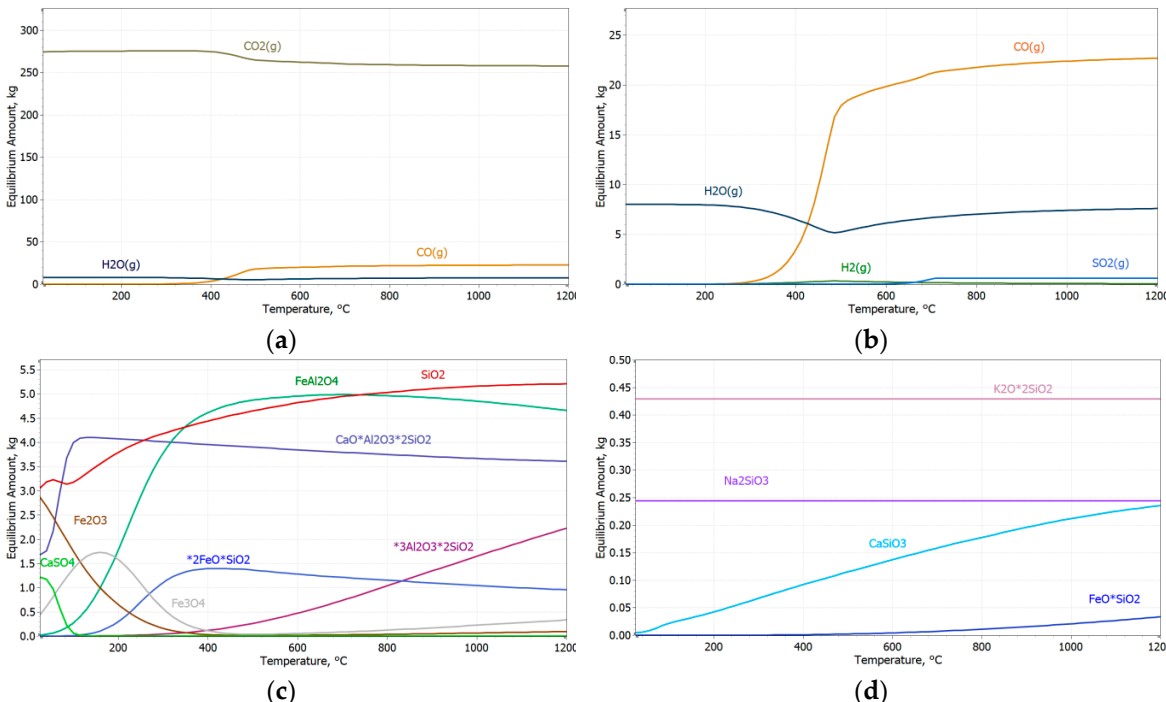

**Figure 5.** Gibbs equilibrium diagrams of coke breeze combustion (calculation for 100 kg of fuel). (**a**) Results for majority gas compounds; (**b**) results for minority gas compounds; (**c**) results for majority solid compounds; and (**d**) results for minority solid compounds.

Figure 6 shows the calculated Gibbs equilibrium diagrams of lignin burning during its complete combustion while the majority product was $CO_{2\ (g)}$ with an approximate 60% share in the mixture of process gases (except for nitrogen). The amount $CO_{(g)}$ and $H_2O_{(g)}$ was considerably higher than that of coke breeze combustion and was related to the significantly higher amount of volatile combustible in lignin (67.90 wt%). The content of gaseous components $CH_{4\ (g)}$ and $H_{2\ (g)}$, which were released at the temperatures of thermal decomposition of the fuel (ca 300–600 °C) was higher as well. The majority mineralogical phases formed at temperatures of lignin combustion include quartz ($SiO_2$), wollastonite ($CaSiO_3$), and oligoclase ($CaAl_2Si_2O_8$). The phases of quartz and oligoclase were identified in X-ray diffraction as well [17]. The predicted minority mineralogical phases include fayalite ($Fe_2SiO_4$), hercynite ($FeAl_2O_4$), hematite ($Fe_2O_3$), magnetite ($Fe_3O_4$), and calcite ($CaCO_3$). The phases of hematite and calcite were identified in X-ray diffraction as well [17].

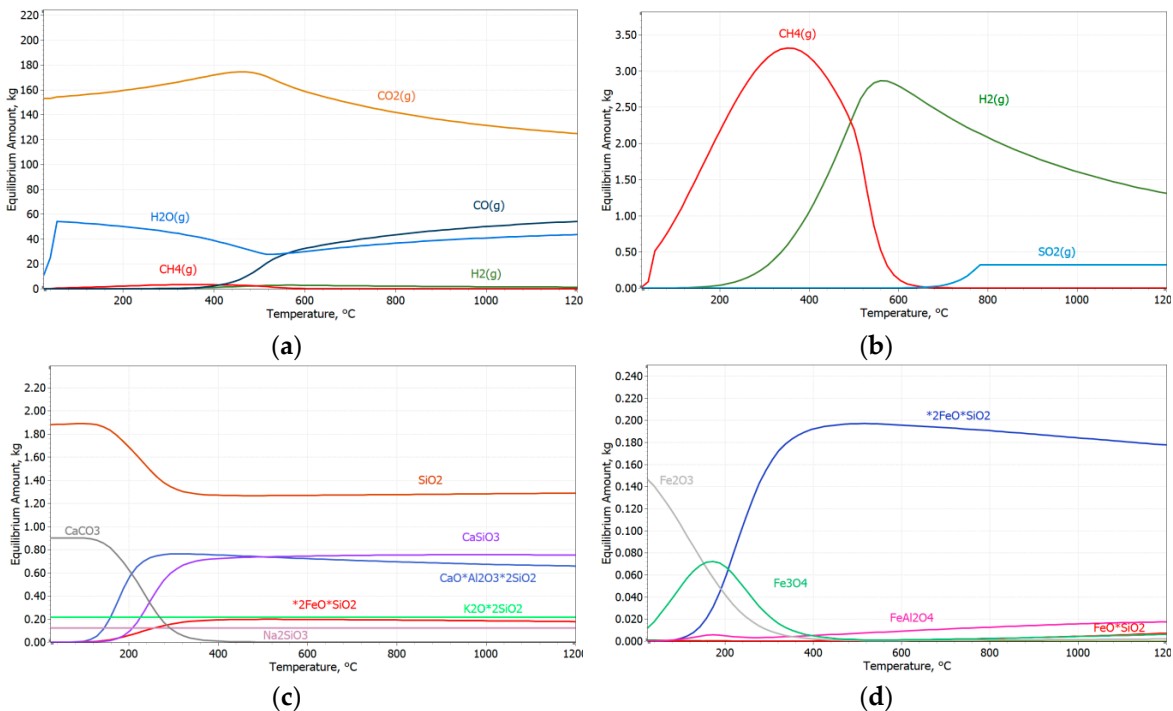

**Figure 6.** Gibbs equilibrium diagrams of lignin combustion (calculation for 100 kg of fuel). (**a**) Results for majority gas compounds; (**b**) results for minority gas compounds; (**c**) results for majority solid compounds; and (**d**) results for minority solid compounds.

Table 4 shows a comparison of fuel ash phase composition in an experimental study by means of X-ray diffraction and the proposed model simulation by means of the HSC program. In the case of both fuels, there was a high match of identified mineral phases. Deviations may be related to the actual preparation of ash for analysis and the kinetic conditions of ash production.

**Table 4.** Comparison of fuel ash phase composition.

| Fuel | Modelling (HSC Program) | | Experimental Analysis | |
|---|---|---|---|---|
| | quartz | $SiO_2$ | quartz | $SiO_2$ |
| | mullite | $Al_6Si_2O_{13}$ | mullite | $Al_6Si_2O_{13}$ |
| Coke | hematite | $Fe_2O_3$ | hematite | $Fe_2O_3$ |
| | hercynite | $FeAl_2O_4$ | anhydrite | $CaSO_4$ |
| | oligoclase | $CaAl_2Si_2O_8$ | augite | $Ca(Mg,Fe)Si_2O_6$ |
| | quartz | $SiO_2$ | quartz | $SiO_2$ |
| | oligoclase | $CaAl_2Si_2O_8$ | oligoclase | $CaAl_2Si_2O_8$ |
| Lignin | calcite | $CaCO_3$ | calcite | $CaCO_3$ |
| | wollastonite | $CaSiO_3$ | hornblende | $(Ca,Na)_2(Mg,Fe,Al)_5$ $(Si,Al)_8O_{22}(OH,F)_2$ |
| | - | - | anhydrite | $CaSO_4$ |

### 3.2. Calculation of Mass and Thermal Balance of Sintering Process with Biomass

In the first step, a mass and thermal balance of coke breeze and biomass (lignin) combustion was determined using the thermodynamic program HSC Chemistry (Heat and Material Balance module). Table 5 gives the mass and thermal balance of the coke combustion process which shows that the calorific value of the coke powder was 28.02 MJ/kg and the amount of ash was 12.43 wt%.

**Table 5.** Mass and thermal balance of coke combustion process (calculation for 1 kg of charge).

| Input Species | Amount (kmol) | Amount (kg) | Amount (Nm$^3$) | Total H (MJ) |
|---|---|---|---|---|
| Coke | 0.07483 | 1.0000 | 0.04930 | −1.45314 |
| C | 0.07110 | 0.854 | 0.00033 | <0.00001 |
| H | 0.00149 | 0.003 | 0.03409 | <0.00001 |
| S | 0.00009 | 0.003 | <0.00001 | <0.00001 |
| O | 0.00019 | 0.006 | 0.00427 | <0.00001 |
| N | 0.00046 | 0.013 | 0.01058 | <0.00001 |
| $Fe_2O_3$ | 0.00021 | 0.0343 | 0.00001 | −0.17677 |
| $SiO_2$ | 0.00073 | 0.0438 | 0.00002 | −0.66399 |
| $Al_2O_3$ | 0.00026 | 0.0266 | 0.00001 | −0.43716 |
| CaO | 0.00015 | 0.0086 | <0.00001 | −0.09737 |
| MgO | 0.00009 | 0.0035 | <0.00001 | −0.05224 |
| $K_2O$ | 0.00002 | 0.002 | <0.00001 | −0.00768 |
| $Na_2O$ | 0.00002 | 0.0014 | <0.00001 | −0.00944 |
| $P_2O_5$ | 0.00001 | 0.0008 | <0.00001 | −0.00848 |
| Air | 0.32428 | 9.3684 | 7.39126 | < 0.00001 |
| $N_{2\,(g)}$ | 0.25294 | 7.08558 | 5.76532 | <0.00001 |
| $O_{2\,(g)}$ | 0.07134 | 2.28282 | 1.62594 | <0.00001 |
| **Output Species** | | | | |
| Process gas | 0.32637 | 10.24714 | 7.43853 | −27.9641 |
| $CO_{2\,(g)}$ | 0.06968 | 3.06658 | 1.58808 | −27.4192 |
| $CO_{(g)}$ | 0.00142 | 0.03983 | 0.03241 | −0.15719 |
| $H_2O_{(g)}$ | 0.00149 | 0.02681 | 0.03336 | −0.3599 |
| $SO_{2\,(g)}$ | 0.00009 | 0.00599 | 0.00213 | −0.02777 |
| $N_{2\,(g)}$ | 0.2534 | 7.09858 | 5.7759 | <0.00001 |
| $O_{2\,(g)}$ | 0.00029 | 0.00935 | 0.00666 | <0.00001 |
| Ash | 0.00081 | 0.12428 | 0.00003 | −1.51264 |
| $Fe_2O_3$ | 0.00017 | 0.02715 | 0.00001 | −0.15637 |
| $2FeO·SiO_2$ | 0.00003 | 0.00558 | <0.00001 | −0.02573 |
| $FeAl_2O_4$ | 0.00005 | 0.00952 | <0.00001 | −0.06941 |
| $CaO·Al_2O_3·2SiO_2$ | 0.00015 | 0.04266 | <0.00001 | −0.64836 |
| $3Al_2O_3·2SiO_2$ | 0.00002 | 0.00877 | 0.00002 | −0.1858 |
| $SiO_2$ | 0.00025 | 0.01505 | <0.00001 | −0.21598 |
| $CaSiO_3$ | 0.00009 | 0.00872 | 0.00001 | −0.13447 |
| $Na_2SiO_3$ | 0.00002 | 0.00276 | <0.00001 | −0.03527 |
| $K_2O·SiO_2$ | 0.00002 | 0.00328 | <0.00001 | −0.03276 |
| $P_2O_5$ | 0.00001 | 0.0008 | <0.00001 | −0.00848 |
| **BALANCE** | **−0.072** | **−0.003** | **−0.002** | **−28.02** |

The experimentally determined parameters of the coke breeze are: Calorific value = 28.16 MJ/kg and amount of ash = 12.1 wt%. Table 6 shows a comparison of the parameters of different fuel combustion, which were used in actual study.

**Table 6.** Parameters of fuel combustion.

| Parameter | Fuel | Modelling HSC Program | Experimental Analysis |
|---|---|---|---|
| **Caloric value** (MJ/kg) | coke 1 | 28.02 | 28.16 |
| | coke 2 | 28.57 | 28.87 |
| | lignin | 22.87 | 23.14 |
| | oak sawdust | 16.43 | 16.56 |
| | pine sawdust | 19.07 | 18.93 |
| | walnut shells | 16.31 | 16.90 |
| | charcoal 1 | 31.86 | 32.66 |
| | charcoal 2 | 29.85 | 29.07 |
| **Ash content** (wt%) | coke 1 | 12.43 | 12.10 |
| | coke 2 | 14.12 | 13.15 |
| | lignin | 3.38 | 3.40 |
| | oak sawdust | 1.59 | 1.50 |
| | pine sawdust | 1.08 | 0.91 |
| | walnut shells | 0.68 | 0.72 |
| | charcoal 1 | 2.33 | 2.30 |
| | charcoal 2 | 4.97 | 5.08 |

It is apparent that the model calculations within the mass and thermal properties of carbonaceous fuel combustion (including biomass) were highly correlated with the experimentally determined properties (see Figure 7).

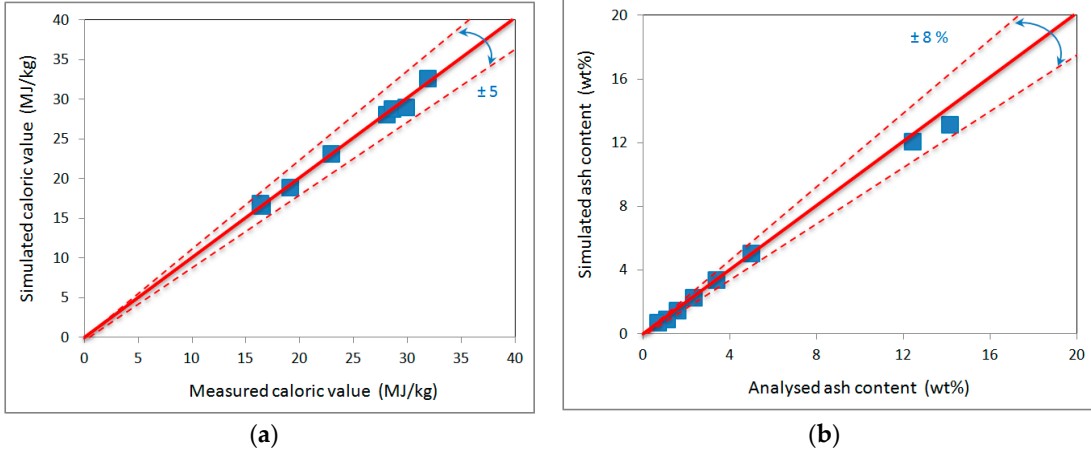

**Figure 7.** Comparison between the measured and simulated results of determining the caloric value (**a**) and ash content in carbonaceous fuel combustion experiments (**b**).

The computational program allows the determination of both the mass and volumetric concentrations of the produced process gas. With the created thermodynamic models, it will be possible to predict the basic parameters for the combustion of any carbonaceous fuels (including biomass) used not only in the sintering process but also in other thermal processes with the reaction of fuel combustion. The developed computational program is thus universally utilisable in various industrial and technical applications.

The sintering process is one of the most ideal thermotechnical processes in metallurgy since the material is heated up to maximum temperatures of approximately 1,450 °C at a relatively low consumption of solid fuel (ca 4–6%). As a result of the heat transfer, the heat produced in the combustion of fuel and the heat supplied in the sintered layer during the ignition of charge is utilised multiple times in the sintered layer. The structure of the input part of the thermal balance of elementary layers with a different distance from the surface of the sintered layer will not be the same. In the upper section of the charge, the elementary layers receive a large portion of heat from the igniting device and the

degree of regeneration is not high. By contrast, the share of regenerated heat in the lower layers is high and the heat received from the igniting device is negligible. The amount of regenerated heat in the zone thermal balance, given by the sum of charge enthalpy (28.96%) and air enthalpy (23.08%), exceeds 50%. The heat transfer in the sintered layer affects the temperature level of the sintering process and thus the completeness of the course of the chemical-mineralogical transformations and the properties of the agglomerate. The effect of heat transfer on the consumption of solid fuel and the performance of the sintering equipment is also substantial. As a result of the heat transfer, the heat produced in the combustion of fuel and the heat supplied by the ignition of charge was utilised multiple times in the sintered layer. The maximum temperature in the sintered layer and the time for which this temperature was maintained in the charge depends both on the amount of heat released by the combustion of solid fuel and the amount of accumulated heat. The maximum temperature in the sintered layer is calculated according to Equation (15):

$$T_{max} = \frac{Q}{m C_b} \tag{15}$$

where $T_{max}$ maximum temperature in the sintered layer (K), $Q$ heat supplied to the elementary layer (kJ), $m$ weight of the elementary layer charge (kg), and $C_b$ specific thermal capacity of agglomeration burden (kJ·kg$^{-1}$·K$^{-1}$).

The fuel combustion zone is limited by the ignition temperature of solid fuel and the temperature of fuel combustion end, at which a permanent drop from the maximum reached temperature occurs. It is apparent that based on the thermal balance that the decisive source of heat for the sintered layer is the heat produced by the combustion of solid fuel [18]. The final structure and composition of agglomerate depends on the conditions of the gas phase flow which provides the supply of oxygen to the combustion zone, the dissociation of charge components, and oxidation-reduction processes, as well as the transfer of thermal energy from the agglomerate cooling zone to the combustion zone. The effect of concentrate/iron ore ratio change on the final agglomerate phase composition is very important [19].

The current mathematical model used in this study was extended with a mass balance of various input components (ferriferous raw materials, basic ingredients, and fuels) while a thermal balance determined on the basis of input and output enthalpies of individual components (including the types of biomass) wasadded to the model as well. The said model allows the control of the overall thermal effect of the sintering process during individual instances of experimental laboratory sintering while it employed the prediction of agglomerate phase composition at the sintering temperatures in the calculations. Since the phase composition of the agglomerate is determined on samples of agglomerate after cooling, the computational model should bring a new perspective of the processes taking place during sintering. The computational model is currently being transformed into software, verified, and adjusted according to the actual outputs from laboratory sintering. The calculation of the thermal balance of sintering process will also be important in terms of determining the energy intensity of the production of iron ore agglomerate. Table 7 shows the mass and thermal balance of agglomerate production using coke breeze.

**Table 7.** Mass and thermal balance of mixture sintering process (calculation for 100 kg of charge).

| Input Species | Temperature (°C) | Amount (kmol) | Amount (kg) | Amount (Nm$^3$) | Total H (MJ) |
|---|---|---|---|---|---|
| Iron concentrate | 25 | 0.310 | 32.977 | 0.006 | −172.39 |
| Iron ore | 25 | 0.252 | 32.960 | 0.007 | −201.07 |
| Dolomite | 25 | 0.044 | 7.952 | <0.001 | −100.49 |
| Calcite | 25 | 0.131 | 13.001 | 0.005 | −157.09 |
| Lime | 25 | 0.036 | 2.000 | 0.001 | −22.75 |
| Coke | 25 | 0.372 | 4.966 | 0.231 | −7.22 |
| Water | 25 | 0.278 | 5.000 | 0.005 | −79.33 |
| Air | 25 | 2.486 | 71.823 | 56.665 | <0.01 |
| **Output Species** | | | | | |
| Process gas | 300 | 2.986 | 90.501 | 67.948 | −263.97 |
| $CO_{2\,(g)}$ | 300 | 0.558 | 24.547 | 12.712 | −212.98 |
| $CO_{(g)}$ | 300 | 0.017 | 0.483 | 0.393 | −1.77 |
| $H_2O_{(g)}$ | 300 | 0.284 | 5.125 | 6.376 | −66.08 |
| $SO_{2\,(g)}$ | 300 | 0.001 | 0.085 | 0.03 | −0.38 |
| $N_{2\,(g)}$ | 300 | 1.941 | 54.383 | 44.25 | 15.7 |
| $O_{2\,(g)}$ | 300 | 0.184 | 5.878 | 4.186 | 1.54 |
| Agglomerate | 1150 | 0.458 | 80.177 | 0.008 | −477.52 |
| FeO | 1150 | 0.058 | 4.157 | 0.001 | −11.72 |
| $Fe_2O_3$ | 1150 | 0.058 | 9.241 | 0.002 | −38.38 |
| $Fe_3O_4$ | 1150 | 0.077 | 17.864 | 0.003 | −68.36 |
| $FeO·SiO_2$ | 1150 | <0.001 | <0.001 | <0.001 | <0.001 |
| $CaO·Fe_2O_3$ | 1150 | 0.18 | 38.747 | <0.001 | −229.61 |
| $MgSiO_3$ | 1150 | 0.048 | 4.775 | 0.001 | −67.34 |
| $SiO_2$ | 1150 | 0.026 | 1.562 | 0.001 | −21.66 |
| $Ca_2SiO_4(L)$ | 1150 | 0.003 | 0.517 | <0.001 | −6.33 |
| $4CaO·Al_2O_3·Fe_2O_3$ | 1150 | 0.006 | 3.062 | <0.001 | −32.11 |
| $Na_2SiO_3$ | 1150 | <0.001 | 0.057 | <0.001 | −0.62 |
| $K_2O·SiO_2$ | 1150 | <0.001 | 0.054 | <0.001 | −0.47 |
| $MnSiO_3$ | 1150 | 0.001 | 0.105 | <0.001 | −0.95 |
| $CaSO_4$ | 1150 | <0.001 | 0.011 | <0.001 | 0.01 |
| $P_2O_5$ | 1150 | 0.001 | 0.025 | <0.001 | 0.02 |
| Balance | - | −0.464 | 0.000 | 11.036 | −1.15 |

The specification of the mass and thermal balance of sintering process is provided in the following points:

(a) The input material composition is based on the actual analysis and actual quantities of materials;

(b) The output quantity and composition of agglomerate are based on the prediction of its mineralogical composition at the sintering temperatures (ca 1000–1350 °C);

(c) The output quantity and composition of sintering gas is calculated based on the prediction of its basic components produced during the fuel combustion at temperatures of approximately 300–1350 °C.

The products' temperatures (process gas and agglomerate) in Table 7 were calculated by minimizing energy in the sintered layer. Table 8 shows the selected parameters of the mass and thermal balance of agglomerate production using coke breeze and various types of biomass while these mathematical models were verified by laboratory experiments. It is apparent that in the event of sintering with certain shares of biomass, it will be necessary to increase the amount in the charge due to the lower calorific values.

**Table 8.** Selected parameters of mass and thermal balance of agglomerate production (calculation for 100 kg of charge).

| Fuel | Amount of Agglomerate (kg) | | Amount of Added Fuel (kg) | | Thermal Efect of Sintering* (MJ/sintering) |
|---|---|---|---|---|---|
| | LSP (real) | HSC (calculated) | LSP (real) | HSC (calculated) | |
| Coke | 78.55 | 80.18 | 4.35 | 4.97 | −1.15 |
| Coke + 20% lignin | - | 77.92 | - | 5.00 | 32.45 |
| Coke + 20% lignin | 76.21 | 77.89 | 5.52 | 5.70 | −1.84 |
| Coke + 20% sawdust 1 | - | 77.92 | - | 5.00 | 48.21 |
| Coke + 20% sawdust 1 | 73.52 | 74.20 | 6.21 | 6.45 | −1.84 |
| Coke + 44% sawdust 1 | 71.44 | 72.81 | 7.04 | 7.85 | −0.95 |
| Coke + 20% sawdust 2 | 71.86 | 72.55 | 8.60 | 8.05 | −1.84 |
| Charcoal | 80.05 | 81.77 | 4.42 | 5.00 | −8.19 |
| Charcoal | 80.15 | 81.35 | 4.29 | 4.72 | −1.19 |

Legend: *—negative thermal efect means sufficiency or excess of heat on sintering.

It is apparent that the model calculations of added fuel are highly correlated with the experimentally determined values (see Figure 8). An even higher correlation was found for the amount of produced agglomerate (simulated and real). Figure 9 shows a comparison of the maximum temperatures in the sintered layer using different biomass coke substitutes. Maximum temperatures are shown illustratively without values and are generalized based on the experimental experience of the authors. The maximum temperatures in the agglomeration process were lower with biomass than with actual coke breeze. Biomass fuels can burn more quickly than coke breeze due to their high porosity and large interface area, while there is a significant increase in the vertical speed of sintering. Lower temperatures in the sintered layer observed with the addition of biomass could also be attributed to the condensation of semi-volatile and volatile organic compounds. These compounds could eventually reduce the heat transfer in the direction of burning. Using biomass up to 20% coke substitution, agglomerates with minimal variations within the chemical, and mineralogical composition were produced. There were larger differences with a substitution of more than 20%.

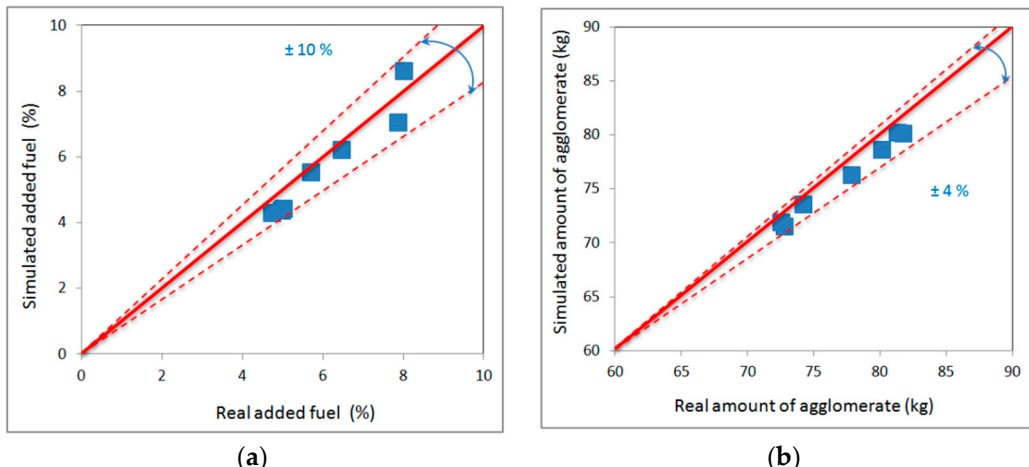

(a)        (b)

**Figure 8.** Comparison between the real and simulated added fuel for sintering (**a**) and real and simulated amount of the agglomerate (**b**).

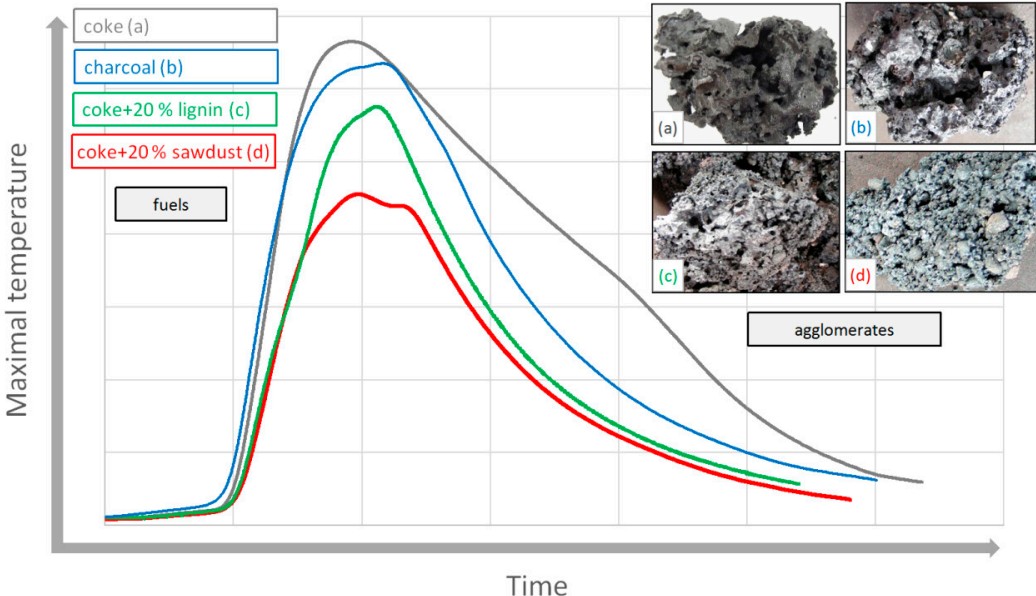

**Figure 9.** Diagram of maximal temperatures in sintering experiments with various fuels. (**a**) Agglomerate produced with coke; (**b**) agglomerate produced with charcoal; (**c**) agglomerate produced with coke + 20% lignin; and (**d**) agglomerate produced with coke + 20% sawdust.

## 4. Conclusions

The aim of this paper was to get the simplest approach (using software HSC Chemistry) to allow one to predict the output parameters based on the initial composition of entered materials. On the basis of the results, the correlation was achieved with the comparison of some values of combustion and sintering during modelling with HSC Chemistry software and experimental simulation. High correlations were found in the identified mineral phases of ash after the combustion of fuels, determining of caloric value and amount of ash in fuels, added fuel for sintering, and determined amount of agglomerate. Some conclusions drawn from thermodynamics modelling were:

1. The majorite phases in ash from coke combustion were quartz and hercynite, majorite phases in ash from lignine combustion were quartz and wollastonite;
2. The calorific value of the biomass used (excluding charcoal) was about 60–80% of the calorific value of coke;
3. Individual types of biomass had (compared to coke) a significantly lower content of ash;
4. In the process of sintering with biomass, it was necessary to increase the amount of total fuel in the charge due to its lower calorific values;
5. The maximum temperatures in the sintering process were lower (about 100–150 °C) with biomass than with coke.

This paper aimed to describe the aplication model that could be used to predict the thermodynamic conditions of carbonaceous fuel combustion (including biomass) in the sintering layer. Pursuant to the modelling of the sintering process in the laboratory conditions, it was feasible to specify the created model by the following parameters:

(a) Calculation of the calorific value of fuels;
(b) Prediction of the phase composition of ash;
(c) Calculation of the quantity of agglomerate;
(d) Prediction of the mineralogical composition of agglomerate at the sintering temperatures;
(e) Calculation of the mass and thermal balance on the basis of the input and output enthalpies of individual components;

(f)    Calculation of the content of Fe$_{TOT}$ in the agglocharge and in the agglomerate pursuant to the stoichiometric conversion of the mineralogical composition.

The created model is comprehensive and might be utilised to unify several computational, monitoring, and evaluation tools (e.g., mass and thermal balance, thermodynamic predictions, monitoring of temperatures and flue gas, determination of reaction mechanism, analysis of agglomerate properties, environmental intensity of the production-amount of $CO_2$ and CO, etc.) into a single interactive model-simulation. The benefit of the new computational model is the possibility to predict the mineral phases of the product during the actual sintering process—high-temperature sintering—which is not in use in the world yet. The new computational model in HSC Chemistry also allowed heat and material balance calculations to be made more easily and faster than any manual method. The developed computational program is universally utilisable in various academic, industrial, and technical applications. The predictions based on the mass and thermal balance start from the thermodynamic calculations. Due to the nature of these calculations, not all important aspects of the sintering operation could be simulated. For the actual sintering, the kinetics of combustion, the kinetics of oxidation-reduction reactions, and the actual physical state of the burden need to be taken into account as well. For this reason, it will also be required to optimise the computational program pursuant to the results of the kinetics experiments.

**Author Contributions:** J.L., M.F., R.F., and M.D. performed experimental analysis; J.L. carried out the investigation, conceptualization, writing and original draft preparation, writing, review, and editing; M.F. evaluated results, formal analysis; R.F. carried out investigation, validation; M.D. performed visualization, project administration.

**Funding:** This research was funded by [APVV] Slovak Research and Development Agency, Slovak Republic number APVV–16-0513.

**Acknowledgments:** Jaroslav Legemza highly appreciates the kind assistance of USS Košice, Slovakia for XRF and XRD of materials used for experiments and Jozef Veselský for translation.

**Conflicts of Interest:** The authors declare no conflict of interest.

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
