# Peer review of "Modelling of Mass and Thermal Balance and Simulation of Iron Sintering Process with Biomass"

_metals, doi:10.3390/met9091010_

Round 1

Reviewer 1 Report

This article presents a "mathematical model that predicts thermodynamic conditions of carbonaceous fuel combustion".

However, there is no mathematical model presented in this paper. As far as I understood, simple mass and heat balances are coupled with piece of thermodynamic software which is simply used to calculate the composition at the equilibrium (in the thermodynamic sense), using as an input the composition obtained by experimental analysis. No kinetic law is taken into account.

Hence, this is clearly a technological paper and I am not convinced that there is an academic interest in this study, even though this kind of approach could be in the major industry interest, as claimed by the authors.

In all cases, a few points should be clarified before the paper be considered for publication:

First of all, the Materials and Method section has to be completed. It is expected to present a mathematical model and no equation are presented here! The mathematical modeling has to be described accurately in this section. Following the previous comment, it is very difficult to get a global view of the method. Please, add at least a scheme presenting the global method with input, output, calculations performed at each step, and present the main assumptions.  Please, if possible use reference in English to allow the reader a better understanding of the sources In eq 10, use T rather than t. Moreover, looking at the equation, I guess that T corresponds to a variation of temperature with respect to initial (room?) temperature and is not an absolute "maximum temperature". Finally choose between °C and °K for the unit but avoid using both. I think that the title and abstract of this paper does not match the content. Reading this, I first thought that the effort was put on numerical approach or mathematical modeling. It should be clearly stated that the aim is to get the simplest  approach (using commercial software to calculate equilibrium) which allows one to predict the output content based on initial composition analysis.

Reviewer 2 Report

The topic of the paper is very interesting, and the proposed model is highly applicable in the industry. Authors not only develop a mathematical model but also validate it against empirical measurements.

The manuscript must be carefully re-read as there are some too long sentences and some sentences difficult to be understand. Here are some comments and suggestions.

The word “the” is too used in the manuscript and in many cases, it is not rightly written. Check the manuscript carefully. This error is typical from some non-english authors.

Lines 32-35. A 4 lines sentences is too long. Split it into several short sentences.

Lines 39-41: This sentence is not clear and very confusing. Please rewrite it.

Lines 46-48: Try to write this sentence in a clearer way. It is confusing.

Line 103: Replace the “value of 14654” from “value from 14654”.

Lines 105-107: Make this sentence clearer. It is hard to understand it.

Lines 128-130: Rewrite this paragraph, as it is har to understand by readers.

Line 142: Replace “layer brings the combustion of fuel.” with “layer is due to fuel combustion”.

Line 148: Replace “of world models shows, there are separate models…” by “The lieterature review shows that there are several separate models…”.

Line 155: Replace “This paper specifies…” by “This paper proposes a mathematical model and a physical….”.

Line 158: Include a reference to the software used.

Line 177: Replace “are presented in another paper of authors” by “are presented in the literature”

Line 233: You talk about the comparison between X-Ray measurements and “a model simulation”. If this model was developed by other authors cite them but if this is the proposed model use “the proposed model” instead “a model”.

Line 247: What units correspond to “Amount”?? (Nm^3)??

Conclusion agree with results obtained along the manuscript.

Round 2

Reviewer 1 Report

The description of calculation has been completed, then I think that Figure 3. particularly helps the reader to understand the method and make the paper much clearer. Thus, the corrections made by the authors meet most of my requirement and i suggest now that the paper be now accepted for publication.

Author Response

Thank you again for your valuable comments and advice on mathematical modeling. A more detailed the description of calculation now specifies the article clearer.